# Expression Profiles of Genes Related to Development and Progression of Endometriosis and Their Association with Paraben and Benzophenone Exposure

**DOI:** 10.3390/ijms242316678

**Published:** 2023-11-23

**Authors:** Francisco M. Peinado, Alicia Olivas-Martínez, Inmaculada Lendínez, Luz M. Iribarne-Durán, Josefa León, Mariana F. Fernández, Rafael Sotelo, Fernando Vela-Soria, Nicolás Olea, Carmen Freire, Olga Ocón-Hernández, Francisco Artacho-Cordón

**Affiliations:** 1Instituto de Investigación Biosanitaria de Granada (ibs.GRANADA), 18012 Granada, Spain; aolivas@ugr.es (A.O.-M.); nolea@ugr.es (N.O.); ooconh@ugr.es (O.O.-H.); 2Centre for Biomedical Research, University of Granada, 18016 Granada, Spain; 3General Surgery, San Cecilio University Hospital, 18016 Granada, Spain; 4Digestive Medicine Unit, San Cecilio University Hospital, 18012 Granada, Spain; 5CIBER Hepatic and Digestive Diseases (CIBEREHD), 28029 Madrid, Spain; 6CIBER Epidemiology and Public Health (CIBERESP), 28029 Madrid, Spain; 7Radiology and Physical Medicine Department, University of Granada, 18016 Granada, Spain; 8Gynecology and Obstetrics Unit, San Cecilio University Hospital, 18016 Granada, Spain; 9Nuclear Medicine Unit, San Cecilio University Hospital, 18016 Granada, Spain; 10Legal Medicine, Toxicology and Physical Anthropology Department, University of Granada, 18071 Granada, Spain

**Keywords:** cell adhesion, metastasis, inflammation, angiogenesis, hormonal stimulation, women, exposure, endocrine-disrupting chemicals

## Abstract

Increasing evidence has been published over recent years on the implication of endocrine-disrupting chemicals (EDCs), including parabens and benzophenones in the pathogenesis and pathophysiology of endometriosis. However, to the best of our knowledge, no study has been published on the ways in which exposure to EDCs might affect cell-signaling pathways related to endometriosis. We aimed to describe the endometriotic tissue expression profile of a panel of 23 genes related to crucial cell-signaling pathways for the development and progression of endometriosis (cell adhesion, invasion/migration, inflammation, angiogenesis, and cell proliferation/hormone stimulation) and explore its relationship with the exposure of patients to parabens (PBs) and benzophenones (BPs). This cross-sectional study included a subsample of 33 women with endometriosis from the EndEA study, measuring their endometriotic tissue expressions of 23 genes, while urinary concentrations of methyl-, ethyl-, propyl-, butyl-paraben, benzophenone-1, benzophenone-3, and 4-hydroxybenzophenone were determined in 22 women. Spearman’s correlations test and linear and logistic regression analyses were performed. The expression of 52.2% of studied genes was observed in >75% of endometriotic tissue samples and the expression of 17.4% (n = 4) of them in 50–75%. Exposure to certain PB and BP congeners was positively associated with the expression of key genes for the development and proliferation of endometriosis. Genes related to the development and progression of endometriosis were expressed in most endometriotic tissue samples studied, suggesting that exposure of women to PBs and BPs may be associated with the altered expression profile of genes related to cellular pathways involved in the development of endometriosis.

## 1. Introduction

Endometriosis is a gynecological disease characterized by the presence of endometrial-like tissue outside the uterine cavity, primarily in the abdominopelvic cavity (peritoneum, ovaries, or rectovaginal septum) [1,2]. This functionally active ectopic tissue is sensitive to hormonal stimulation and can originate cyclic bleeding, promoting the appearance of local inflammatory reactions and triggering pain (dysmenorrhea, dyspareunia, or chronic pelvic pain), gastrointestinal disorders, or infertility, among other maladies [3,4].

The pathogenesis and pathophysiology of endometriosis have yet to be fully elucidated, although various theories have been proposed. Increasing evidence has been published over recent years on the implication of endocrine-disrupting chemicals (EDCs) [5] in an increased risk of endometriosis, which has been associated with human exposure to several families of EDCs [6,7,8], including parabens (PBs) and benzophenones (BPs) [6]. PBs, which are widely used as preservatives in personal care products (PCPs), pharmaceuticals, and food [9], include methyl- (MeP), ethyl- (EtP), propyl- (PrP), and butyl-paraben (BuP) congeners. BPs, which frequently serve as UV-filters in PCPs [10], include benzophenone-1 (BP-1), benzophenone-3 (BP-3), and 4-hydroxybenzophenone (4-OHBP) congeners.

In vitro and in vivo studies have demonstrated the estrogen-like effects of PBs and BPs, supporting the hypothesis that they promote endometriosis, an estrogen-dependent disease [11,12]. EDC exposure has also been associated with inflammation and oxidative stress [13,14], although its role in the pathophysiology remains unknown. Five key steps in the development and progression of endometriotic lesions—cell adhesion to the peritoneum; invasiveness into the mesothelium; recruitment of inflammatory cells; angiogenesis in endometriotic tissue; and cell proliferation—have been previously described [15]. Changes in mediators of the corresponding cell-signaling pathways have been observed in studies of endometriosis [16,17,18,19,20,21,22,23,24,25]. Thus, in comparison to normal endometrial tissue, ectopic tissue has evidenced upregulation of the following: integrins and claudins, key members of tight junctions [16,17]; metalloproteases (*MMPs*), involved in extracellular matrix cleavage and therefore invasiveness [18,19]; proinflammatory interleukins [20,21]; certain angiogenesis-related genes [22,23]; and genes related to cell proliferation [24,25]. Currently, the identification of different biomarkers of endometriosis is of crucial importance due to the need for non-invasive diagnostic methods of the disease [26]. Additionally, these biomarkers could predict whether women will respond to first-line treatment for this disease. To our best knowledge, no study has been published on the ways in which exposure to EDCs might affect these cell-signaling pathways. Following the observation by our group that exposure to PBs and BPs is associated with endometriosis risk [6], the present study was designed to determine the expression profile of a panel of 23 genes related to five key cell-signaling pathways for endometriosis development (cell adhesion; invasion, migration and metastasis; inflammation; angiogenesis; and cell proliferation and hormone stimulation) in women with endometriosis and to explore its relationship with their exposure to PBs and BPs. These families of EDCs were selected in this study due to (i) the growing evidence of the relationship between the consumption of cosmetics and PCPs, human exposure to PBs and BPs, and adverse health effects; (ii) previous studies reporting positive associations between PB/BP exposure and endometriosis risk [6,8]; and (iii) the scarcity of information on PB/BP exposure-related adverse outcome pathways in endometriosis.

## 2. Results

### 2.1. Characteristics of the Study Population and Urinary PB and BP Concentrations

The sociodemographic and reproductive characteristics of all participants (n = 33) are summarized in Appendix A. Out of the 33 women in the study, an adequate sample for exposure and gene expression measurements was obtained from 22, whose characteristics are reported in Appendix A. The mean age of the entire study participants (n = 33) was 38.0 ± 7.3 years; 60.6% (n = 20) were in a normal weight range (BMI < 25 kg/m^2^), 57.6% (n = 19) lived in a rural area, 63.6% (n = 21) did not have a university degree, 72.7% (n = 24) were employed outside the home, 45.5% (n = 15) were nulliparous, 63.6% (n = 21) had moderate/severe menstrual bleeding, and 75.8% (n = 25) had a diagnosis of ovarian/peritoneal endometriosis, with 63.6% (n = 21) being in stage I/II (Appendix A).

Our group previously reported the urinary PB and BP concentrations obtained for participants in the EndEA study [6]. All compounds were detected in all samples from our subsample of 22 women (Appendix A). The PB and BP congeners with the highest concentrations were MeP and BP-3, respectively.

### 2.2. Gene Expression Levels and Associations with PB and BP Concentrations

Gene expression levels in the endometriotic tissues from the entire cohort are exhibited in Table 1, showing that 12 genes (52.2%) were expressed in >75% of samples and 4 (17.4%) in 50–75%. Both genes related to cell adhesion were expressed in all samples, whereas only four of the seven invasion-, migration-, and metastasis-related genes (*MMP1*, *RRM2*, *RHOB*, and *SPRY2*) and three of the five inflammation-related genes (*IL1RL1*, *IL6ST*, and *NR3C1*) were expressed in more than half of samples. Finally, all angiogenesis-related genes and three of the genes related to cell proliferation and hormonal stimulation (*DUSP6*, *ERα* and *STAR*) were expressed in more than half of samples. The sole difference in expression between endometriosis stages was observed for two genes related to the invasion, migration, and metastasis pathway (*FUT8* and *SPRY2* genes), which showed higher gene expression levels in patients with stages III/IV. In addition, a higher close-to-significant expression of *DUSP6* (*p*-value = 0.092) was observed in patients with stages III/IV (Appendix A). Appendix A summarizes the gene expression profile of the subset of women with assessed values of both paraben/benzophenone exposure and gene expression levels (n = 22).

Spearman correlation coefficients between PBs/BPs and gene expression levels (n = 22) are displayed in Appendix A. Associations between PB/BP exposure and gene expression levels (n = 22) are reported in Table 2, Table 3, Table 4, Table 5 and Table 6. Significant associations were also shown in dot plots (genes expressed in >75% of samples) and box plots (genes expressed in 25–75% of samples) (Appendix A). The concentration of at least one PB congener was associated with the expression of gene(s) involved in each of the five pathways under study, while BPs were associated with genes related to invasion, migration/metastasis, inflammation, cell proliferation, and hormonal stimulation cell-signaling pathways but not the adhesion and angiogenesis pathways.

#### 2.2.1. Markers of Cell Adhesion

Urinary concentrations of MeP, BuP, and ∑PBs showed significant positive correlations with ectopic tissue *ITGB2* and *CLDN7* gene expression (Table 2 and Appendix A). When concentrations were considered as dichotomous variables (higher vs. lower), expression of *ITGB2* was related to higher concentrations of MeP, BuP, and ∑PBs, and its positive association with EtP was close to statistical significance (*p*-value = 0.091). A similar relationship was observed between ectopic expression of CLDN7 and higher concentrations of MeP, BuP, and ∑PBs. Furthermore, CLDN7 gene expression was greater for the third exposure tertile of both BuP and ΣPBs, with statistical significance in the case of BuP and almost reaching statistical significance in the case of ΣPBs (*p*-value = 0.066) (Appendix A). No significant associations were found between BP exposure and *ITGB2* or *CLDN7* gene expression levels.

#### 2.2.2. Markers of Invasion, Migration, and Metastasis

Urinary MeP and ∑PBs concentrations were positively associated with the expression of *MMP1* and *RHOB* genes, and EtP and BuP concentrations were also positively related to the expression of *RHOB* gene (Table 3 and Appendix A). Similar associations were observed when exposure was dichotomized, with increased risk of detectable *MMP1* and *RHOB* expression levels in patients with higher exposure to MeP and ∑PBs and of detectable *RHOB* expression in patients with higher exposure to BuP. In addition, *RHOB* gene expression levels gradually increased when BuP was categorized into tertiles. A similar trend was observed for tertiles of ∑PBs and *MMP1* (Appendix A). Moreover, detectable expression of *FUT8* was observed in the women with higher exposure to PrP, while the higher expression levels of RRM2 in women with a higher concentration of MeP was close to statistically significant (*p*-value = 0.088). By contrast, MeP and ∑PBs concentrations were inversely associated with the detectable expression of *SPRY2*. A positive association was found between concentrations of ∑BPs and detectable expression of *MMP1*. *MMP7* expression was detected in <25% of the study population; therefore, no linear or logistic regression was performed.

#### 2.2.3. Markers of Inflammation

MeP, BuP, and ∑PB concentrations were significantly correlated with *NR3C1* and *IL6ST* expression levels, and higher MeP, BuP, and ∑PB concentrations were also correlated with *IL6ST* expression levels (Table 4 and Appendix A). Women with higher concentrations of BuP also showed increased *IL1RL1* expression, although statistical significance was not reached (*p*-value = 0.055). Moreover, the expression of *IL6ST* and *NR3C1* gradually increased with higher tertiles of BuP concentrations (Appendix A). *TNFRSF1B* expression was positively associated with urinary concentrations of BP3 and ƩBPs, and a similar association was observed between *TNFRSF1B* and concentrations of BP3 and ƩBPs when the exposure was dichotomized, although the latter did not reach the statistical significance (*p*-value = 0.080). Higher *IL1RL1* levels were also observed in the participants with higher 4-OHBP concentrations. *IL1R2* expression was detected in <25% of the study population; therefore, no linear or logistic regression was performed.

Urinary concentrations of MeP and ∑PBs were significantly correlated with *ANGPT1* exposure, both as a continuous and dichotomous variable (Table 5 and Appendix A). MeP and ∑PBs were close to significantly associated with *ANG* expression levels (*p*-values = 0.062 and 0.063, respectively). PrP concentrations were positively associated with the expression of *sVEGFR-1*, and BuP showed a similar association when considered as a dichotomous variable. Finally, reduced *VEGFA* expression was observed in the women with higher EtP concentration, although statistical significance was not reached (*p*-value = 0.080). No association was observed between concentrations of BP congeners and the expression of angiogenesis-related genes, except for lower *VEGFA* expression in the women with higher concentrations of BP-1 and BP-3, although these associations did not reach statistical significance (*p*-values = 0.080 in both cases).

*DUSP6* gene expression was inversely associated with concentrations of MeP and ƩPBs. *ERα* expression was significantly and positively related to MeP and ƩPBs concentrations, both as a continuous and dichotomous variable, and was also positively and close-to-significantly associated with EtP and BuP (*p*-values = 0.074 and 0.076, respectively). Lower *STAR* gene expression was associated with higher MeP concentrations, although statistical significance was not reached when considered as a continuous variable (*p*-value = 0.086), while *STAR* levels were positively associated with concentrations of BP-1 and 4-OHBP (Table 6 and Appendix A). Expression of *CYP19A1* and *PGR* was detected in <25% of the study population; therefore, no linear or logistic regression was performed.

## 3. Discussion

To the best of our knowledge, this is among the very first studies to describe the expression profile of key genes involved in the main molecular processes related to endometriosis in human endometriotic tissue and its potential association with exposure to EDCs. Most of the studied genes were expressed in the majority of samples, and exposure to certain PB and BP congeners was positively associated with the expression of genes involved in cell adhesion; invasion, migration, and metastasis; inflammation; angiogenesis; and cell proliferation and hormonal stimulation, critical molecular processes for the onset and progression of this disease. This evidence should be added to our previous report addressing the associations between PB/BP exposure and expression profiles of genes related to other cell molecular processes related to endometriosis, such as the cell cycle, cell differentiation, and lipid metabolism [27].

Adhesion of endometrial cells to the peritoneal surface is considered a primary hallmark of endometriosis in its initial stages [28]. Exposure to various PBs was associated with adhesion-related *ITGB2* and *CLDN7* genes in the present study, in line with recent in vitro findings of enhanced adhesive capacity in human umbilical vein endothelial cells after exposure to different EDCs [29]. Upregulation of *CLDN7* has also been reported in ovarian cancer cell lines [30], although the evidence is contradictory, given that downregulation of *CLDN7* has been observed in human endometriotic lesions [31] and endometrial cancer, promoting proliferation and metastasis [32].

The adhesion of endometrial cells to the peritoneal surface is followed by a process of invasion, migration, and metastasis that involves the overexpression of metalloproteinases [33]. *MMP1* was expressed in most of the present endometriotic samples, in agreement with previous reports of *MMP* overexpression in women with endometriosis [34,35] and more severe stages of this disease [19,36]. An increase in *MMP* activation has been detected in ovarian cancer, enhancing cell invasiveness [30]. In the present investigation, *MMP1* expression was positively associated with exposure to MeP, ∑PBs, and ƩBPs, which may therefore favor the invasion, migration, and metastasis of endometriotic cells. In vitro studies have linked PB exposure to increased *MMP* expression and activity in dermal fibroblasts [37] and breast cancer cells [38]. Exposure to other EDCs also increased *MMP* expression in ovarian cancer cell lines [39]. *RHOB*, *FUT8*, and *RRM2* genes may also participate in invasion, migration, and metastasis. The expression of RHOB has not been described in endometriosis, but upregulation of this gene has been reported in human breast tumors [40]. The present study describes for the first time the presence of *RHOB* in endometriotic tissue and its positive association with exposure to most PB congeners studied. This suggests that exposure to these EDCs may contribute to the invasiveness of endometriotic tissue in a similar manner to the observed effects of EDCs on metastasis in breast cancer. *FUT8* has not previously been studied in endometriosis tissue samples but has been implicated in the epithelial–mesenchymal transition and metastasis [41,42]. PrP concentrations were positively associated with *FUT8* expression, suggesting that exposure to PrP may increase endometriotic tissue *FUT8* and promote metastasis and progression to more advanced stages of disease. It is also suspected that *RRM2* upregulation may promote cell invasion and metastasis in cancer and ovarian endometriosis [43,44]. Its expression was increased in the present women with higher MeP exposure, which may therefore play a potential role in boosting invasiveness. Finally, although the importance of *SPRY2* in endometriosis is unknown, it plays a crucial role in the regulation of cancer cell invasion [45,46]. According to the present findings, MeP and ƩPBs might downregulate *SPRY2* expression and thereby favor invasiveness and metastasis in women with endometriosis.

Another hallmark of endometriosis is a complex proinflammatory microenvironment, which favors cellular adhesion and invasion processes and the vascularization and proliferation of endometriotic lesions [47]. Upregulation of *IL1RL1*, *IL6ST*, *TNFRSF1B*, and *NR3C1* has been described in women with endometriosis [48,49]. In line with the proposal that human exposure to PBs/BPs may be related to inflammation [14,50], the present results suggest that the upregulation of *IL1RL1*, *IL6ST*, *TNFRSF1B*, and *NR3C1* genes associated with exposure to various congeners of PBs and BPs could contribute to the generation of an inflammatory environment in endometriotic tissue.

Angiogenesis plays an important role in the pathogenesis and pathophysiology of endometriosis, and various genes are involved in this molecular process [51]. Increased *ANG*, *VEGFA*, and *sVEGFR-1* expression was previously observed in the peritoneal fluid of women with endometriosis [23,52]. In the present study, urinary concentrations of various PB congeners were positively associated with *ANG*, *ANGPT1*, and *sVEGFR-1* expression. Despite associations not reaching statistical significance, exposure to BP-1 and BP-3 was unexpectedly related to downregulated levels of VEGFA. Although they were in accordance with a previous study carried out in Ehrlich ascites carcinoma and Dalton’s lymphoma ascites cells [53], others have found upregulation of VEGFA after exposure to other EDCs such as bisphenols in breast cancer cells [54] and upregulation of VEGFD in endometrial cells [55], indicating that our non-significant associations between BPs and VEGFA might be spurious. The present findings support the hypothesis that exposure to PBs could favor the development of pro-angiogenic properties via endometriotic lesions, as suggested by the observation of increased microvessel density after exposure to bisphenol A in a model of neuroblastoma [56].

Estradiol (E_2_) promotes the persistence, multiplication, and progression of endometriotic lesions, acting as a growth factor for this tissue [57], and endometriosis is closely associated with steroid metabolism and associated pathways [3,58]. E_2_ production requires the action of *STAR*, which facilitates the entry of cytosolic cholesterol into the mitochondrion, and *CYP19A1*, which is responsible for aromatization of androgens into estrogens [59,60]. In the present study, *STAR* gene was expressed in most of the ectopic tissue samples, and exposure to BP-1 and 4-OHBP was positively associated with STAR expression, indicating that exposure to these EDCs might be related to increased E_2_ production in endometriotic tissue. These results support the previous suggestion of a potential role for EDCs in the upregulation of *CYP19A1* [61]. In contrast, a reduction in *STAR* expression was observed in the women with higher concentrations of MeP. Endometriotic lesions are very sensitive not only to the production of E_2_ but also to the presence of circulating estrogens. In this regard, E_2_ membrane receptors (*ERα* and *ERβ*) have been widely reported in endometriotic tissue [62], with some researchers describing *ERβ* as the predominant receptor on the cell surface [63]. Hence, although *ERβ* gene expression was not evaluated, *ERα* gene expression in endometriotic tissue was found to be positively related to exposure to most PB congeners examined, indicating that this exposure might contribute to the sensitivity of this tissue to circulating E_2_. The influence of BPs or PBs on ERα expression has not previously been studied, but the present results are in line with in vivo findings of *ERα* overexpression after exposure to BPA [64].

*DUSP6* gene encodes a protein that counteracts cellular proliferation and is dysregulated in various diseases, including cancer. The number of cells expressing *DUSP6* was reported to be lower in ectopic versus eutopic endometrium [65], suggesting this gene might also play a crucial role alongside E_2_ in enhancing cell proliferation in endometriotic tissue. In the present investigation, MeP and ΣPBs were associated with a lower expression of this gene in endometriotic tissue, suggesting that these EDCs might contribute to the downregulation of *DUSP6* in this tissue.

Study limitations include the small sample size, reducing the statistical power, although it is highly challenging to gather endometriotic tissue samples for investigation. The limited amount of endometriotic tissue sample available prevented us from performing protein validation analysis to confirm the reported gene expression results. Furthermore, the utilization of spot urine samples prevented consideration of the variability in daily exposure to analytes with a relatively short elimination half-life. Nevertheless, samples were all first-morning urine samples taken during hospitalization before endometriosis surgery. In addition, only two families of EDCs were measured, with no evaluation of the combined effect of PBs and BPs alongside other EDCs. Furthermore, given the reported estrogenic/antiandrogenic effect of these compounds and possible similar underlying mechanisms of action [11,12], it is not possible to determine whether individual differences in these compounds could be the cause of the different associations found between EDC concentrations and gene expression levels. Finally, the ability to include covariates in regression analyses was limited by the small study population. Study strengths include the measurement, for the first time in endometriotic tissue, of numerous genes involved in cellular pathways related to the pathophysiology of endometriosis. Importantly, many cases revealed consistent associations between gene expression and exposure to certain PB/BP congeners. The combined investigation of biomarkers of exposure and potential biomarkers of effect yielded evidence of different pathways for adverse outcomes in endometriosis.

## 4. Material and Methods

### 4.1. Study Population and Sample Collection

This cross-sectional study, which forms part of the hospital-based case-control EndEA study, included 33 women with endometriosis undergoing surgery at the Gynecology and Obstetrics Units of San Cecilio and Virgen de las Nieves University Hospitals (Granada, Southern Spain) from January 2018 to July 2019 [6,7,27]. All cases were diagnosed with endometriosis via laparotomy or laparoscopic surgery and histological confirmation. Inclusion criteria were premenopausal status, age between 20 and 54 years, receipt of abdominal surgery, and body mass index (BMI) below 35 kg/m^2^. Exclusion criteria were history of cancer (except non-melanoma skin cancer), pregnancy at study enrolment, and inability to read and sign the informed consent document. Women were staged according to the Revised American Fertility Society classification [66]. Written informed consent was provided by all participants, and the study was approved by the Research Ethics Committee of Granada (0464-N-18).

Endometriotic tissue samples were gathered during surgery and kept in QIAazol reagent (Qiagen, Hilden, Germany) to ensure RNA stability, and first-morning spot urine samples were collected in fasting conditions (fasting time > 8 h) on the same day as the surgery and were kept in PB- and BP-free glass tubes. All samples were immediately stored at −80 °C at the Biobank of the Public Andalusian Health Care System until their analysis. The BMI of participants was calculated, and data were gathered on sociodemographic, lifestyle, and clinical variables from epidemiological and clinical questionnaires completed by participants and on surgical variables from questionnaires completed by surgeons (Appendix A). Endometriotic tissue samples were available from 33 participants, while urine samples were obtained from only 22 of them.

### 4.2. RNA Isolation and Quantitative Real-Time Polymerase Chain Reaction (qRT-PCR)

Endometriotic tissue samples were maintained in QIAzol reagent (Qiagen, Hilden, Germany) during the homogenization process to ensure RNA stability. Total RNA was extracted from 30 mg of sample with the RNeasy Mini kit (Qiagen, Hilden, Germany) according to the manufacturer’s protocol. Final RNA concentration and quality (260/280 ratio) were determined using a NanoDrop 2000 (Thermo Fisher Scientific, Waltham, MA, USA). Total RNA (1000 ng) was transcribed into cDNA with the iScriptAdvanced cDNA Synthesis Kit for RT-qPCR (Bio-Rad Laboratories, Hercules, CA, USA) according to the manufacturer’s instructions.

Real-time PCR was carried out with a CFX96 Real-time PCR detection system (Bio-Rad Laboratories, Hercules, CA, USA) using SsoAdvanced SYBR^®^ Green Supermix (Bio-Rad Laboratories, Hercules, CA, USA). The manufacturer’s protocol was followed to measure the expression of 23 genes involved in the five cell-signaling pathways: cell adhesion (integrin beta-2 (*ITGB2*) and claudin 7 (*CLDN7*)); migration/invasion (matrix metallopeptidase 1 (*MMP1*), matrix metallopeptidase 7 (*MMP7*), fucosyltransferase 8 (*FUT8*), ribonucleotide reductase M2 (*RRM2*), midkine (*MDK*), ras homolog gene, family, member B (*RHOB*), and sprout homolog 2 (*SPRY2*)); inflammation (interleukin 1 receptor, type I (*IL1R1*), interleukin 1 receptor, type II (*IL1R2*), interleukin 6 cytokine family signal transducer (*IL6ST*), nuclear receptor subfamily 3 group C member 1 (*NR3C1*), and tumor necrosis factor receptor superfamily member 1B (*TNFRSF1B*)); angiogenesis (angiogenin (*ANG*), angiopoietin 1 (*ANGPT1*), soluble vascular endothelial growth factor receptor-1 (*sVEGFR-1*), and vascular endothelial growth factor A (*VEGFA*)); and cell proliferation/hormone stimulation (cytochrome P450 family 19 subfamily A member 1 (*CYP19A1*), dual specificity phosphatase 6 (*DUSP6*), estrogen receptor alpha (*ERα*), progesterone receptor (*PGR*), and steroidogenic acute regulatory protein (*STAR*)). Briefly, 10 μL SsoAdvanced Universal SYBR green mix, 1 μL cDNA (250 ng), and 1 μL of each PCR primer in 5 μmol L^−1^ were mixed together. Nuclease-free water was added to achieve the final volume of 20 μL. qPCR reaction conditions were the following: first, the initiation process took place, for which the temperature reaction reached a temperature of 95 °C for two minutes (1 cycle); subsequently, the denaturation process occurred through 40 cycles of 5 s at a temperature of 95 °C; next, the alignment and extension process took place, through 40 cycles of 60 s at a temperature of 65–95 °C; finally, to evaluate the reaction specificity of the assay, a cycle of 5 sec/step was carried out, at a temperature of 65–95 °C (0.5 °C increments), showing a melt curve for each of the genes in each sample analyzed. Primers used for these studies were purchased from Bio-Rad Laboratories (Hercules, CA, USA), and their assay ID, amplicon context sequence, amplicon length (bp), and chromosome location are detailed in Appendix A. Every mRNA primer used in this study was designed and experimentally tested by Bio-Rad Laboratories, generating an amplification plot using universal RNA, conducting a melt curve analysis, and calculating the efficiency and dynamic range from a seven-point standard curve. Specificity was determined by next-generation sequencing of the amplicon.

The criteria for the selection of these genes was based on previous evidence relating exposure to EDCs with the gene expression of these genes, their participation in the different cell-signaling pathways, and their relationship with endometriosis [67,68].

Gene expression was analyzed via Bio-Rad CFX 96 Manager Software to determine the cycle of quantification (Ct), and this was calculated using the 2^−ΔΔCt^ method [69]. All values were normalized using glyceraldehyde-3-phosphate dehydrogenase (*GAPDH*) RNA expression levels, calculating the difference corresponding to the quantification cycles between the target genes and GAPDH gene: ∆Ct = Ct (gene of interest)—Ct (GAPDH). Amplification efficiencies of the 23 primers used were ranged between 91 and 104%, being similar to that of the reference gene GAPDH (97%).

### 4.3. Chemical Analysis

Dispersive liquid–liquid microextraction (DLLME) and ultra-high-performance liquid chromatography with tandem mass spectrometry (UHPLC-MS/MS) were performed to determine urinary concentrations of MeP, EtP, PrP, BuP, BP-1, BP-3, and 4-OHBP, as previously described [70].

After thawing urine samples at room temperature and centrifuging at 2600× *g* for 10 min, 1.0 mL was taken for analysis. The total amount of these benzophenones and parabens (free and conjugated) was obtained by enzymatically treating samples with an enzyme solution of β-glucuronidase/sulfatase, pre-prepared by dissolving 10 mg of β-glucuronidase/sulfatase (3·10^6^ U g solid^−1^) in 15 mL of 1M ammonium acetate/acetic acid buffer solution (pH 5.0). Enzymatically treated samples were incubated at 37 °C for 24 h. Next, 20 μL of standard replacement solution (5 mg/L of EP^_13^C_6_, 2 mg/L of BPA-D_16_, and 2 mg/L of BP-d_10_) were added, and samples were diluted with 10 mL of 10% aqueous NaCl solution (pH 2.0, adjusted with 0.5 M HCl). Samples were then mixed with a solution of 1 mL of acetone (dispersing solvent) and 0.5 mL of trichloromethane (extraction solvent), shaken manually for 30 s, and centrifuged at 4000× *g* for 10 min. The organic phase was then collected with care from the bottom of the glass tube using a 1 mL pipette and was placed in 2 mL glass vials. All of the extracted fluid was evaporated under a nitrogen stream, and the residue was dissolved with 100 μL of an acetonitrile/water mixture (0.1% ammonia, 70:30 [*v*/*v*]) and vortexed for 30 s. The extract was then ready for analysis via UHPLC-MS/MS. The limit of detection (LOD) was determined as the minimum detectable amount of analyte, with a signal-to-noise ratio ≥ 3. LODs were 0.05 ng/mL for BP-1, 0.06 ng/mL for BP-3 and 4-OHBP, and 0.10 ng/mL for MeP, EtP, PrP, and BuP.

### 4.4. Statistical Analysis

In a descriptive analysis, arithmetic means and standard deviations were calculated for continuous variables and relative frequencies for categorical variables. Urinary concentrations of PBs, BPs, and gene expression levels were summarized as arithmetic means with standard deviations and as 25, 50, and 75 percentiles. The Shapiro–Wilk test was applied to check the normality of variable distributions, and data found to be non-normally distributed underwent logarithmic transformation.

Spearman’s rank correlation coefficient was used to evaluate monotonic correlations between urinary concentrations of PBs/BPs and the expression of genes with detection/expression frequencies above 75%. Simple linear regression models were also developed to evaluate the relationship between PB/BP exposure and gene expression levels, expressing the results as β with 95% confidence intervals. In parallel, simple linear regression models were constructed after patient stratification (i) into lower or higher PB and BP exposure, i.e., below or above the median concentration of each congener; and (ii) into low, moderate, and high exposure, based on tertiles of PB and BP concentrations. Genes expressed in 25–75% of samples were considered as dichotomous variables (detected/not detected), and simple logistic regression models were used to evaluate their association with PBs and BPs (considering assumptions of each regression modality), while no analyses were performed on genes expressed in <25% of samples.

Statistical analysis was performed with SPSS Statistics 23.0 (IBM, Armonk, NY, USA) analysis, and the level of significance was *p* < 0.05 in all tests; however, associations with *p*-values between 0.05 and 0.10 are cautiously considered, given the limited sample size. Due to sample availability, descriptive analysis on sociodemographic/gynecologic characteristics and gene expression profiles were summarized from the whole cohort (n = 33), while urinary concentrations of PBs/BPs and the associations between expression profiles and exposure were accomplished with data from the subset of women with both samples collected (n = 22).

## 5. Conclusions

Genes related to the development and progression of endometriosis were expressed in most of this series of endometriotic tissue samples. The results obtained indicate that the exposure of women to PBs and BPs may be associated with the expression of genes involved in different molecular processes that play key roles in the development of endometriosis, including cell adhesion; invasion, migration, and metastasis; inflammation; angiogenesis; and cell proliferation and hormonal stimulation. Given the novelty of these results, further studies with larger sample sizes are warranted in order to confirm the impact of human exposure to EDCs on the pathophysiology of endometriosis.

## Figures and Tables

**Table 1 ijms-24-16678-t001:** Gene expression levels in endometriotic tissue (n = 33).

Cell Pathway	Gene	n	FD (%)	AE (%)	Mean	St. Dev.	Percentiles		
							25	50	75
Cell adhesion	ITGB2	33	100	98.0	2.37	5.71	0.42	0.85	2.26
	CLDN7	33	100	98.0	1.59	2.72	0.56	0.95	1.62
Invasion, migration and metastasis	MMP1	21	63.6	98.0	10.76	54.08	n.e.	0.03	0.31
	MMP7	4	12.1	97.0	1.01	5.31	n.e.	n.e.	n.e.
	FUT8	11	33.3	98.0	0.73	2.89	n.e.	n.e.	0.01
	RRM2	33	100	96.0	2.79	9.27	0.51	0.91	1.59
	MDK	14	43.4	100.0	0.29	0.94	n.e.	n.e.	0.02
	RHOB	33	100	96.0	5.18	18.49	0.36	0.75	2.09
	SPRY2	21	63.6	91.0	0.42	0.72	n.e.	0.22	0.52
Inflammation	IL1R2	4	12.1	98.0	3.18	10.99	0.04	0.28	0.92
	IL1RL1	27	81.8	98.0	0.30	1.19	n.e.	n.e.	n.e.
	IL6ST	33	100	104.0	7.60	33.93	0.34	0.75	1.62
	NR3C1	30	90.9	99.0	12.08	57.29	0.11	0.53	2.16
	TNFRSF1B	11	33.3	101.0	0.79	3.11	n.e.	n.e.	n.e.
Angiogenesis	ANG	33	100	96.0	2.71	7.76	0.49	0.88	1.98
	ANGPT1	28	84.8	100.0	1.59	3.85	0.04	0.43	1.36
	sVEGFR-1	23	69.7	97.0	5.82	32.28	n.e.	0.07	0.20
	VEGFA	23	69.7	101.0	1.05	1.19	n.e.	0.65	1.56
Cell proliferation and hormonal stimulation	CYP19A1	5	15.2	102.0	2.22	12.01	n.e.	n.e.	n.e.
	DUSP6	31	93.9	99.0	5.91	7.88	0.03	3.00	8.37
	ERα	33	100	97.0	2.25	6.40	0.53	0.97	1.59
	PGR	1	3.0	96.0	0.00	0.01	n.e.	n.e.	n.e.
	STAR	28	84.8	94.0	8.34	27.11	0.46	1.00	2.38

FD: frequency of detection; AE: amplification efficiency; St. Dev.: standard deviation; ITGB2: integrin beta-2; CLDN7: claudin 7; MMP1: matrix metallopeptidase 1; MMP7: matrix metallopeptidase 7; FUT8: fucosyltransferase 8; RRM2: ribonucleotide reductase M2; MDK: midkine; RHOB: ras homolog gene family, member B; SPRY2: sprout homolog 2; IL1RL1: interleukin 1 receptor, type I; IL1R2: interleukin 1 receptor, type II; IL6ST: interleukin 6 cytokine family signal transducer; NR3C1: nuclear receptor subfamily 3 group C member 1; TNFRSF1B: tumor necrosis factor receptor superfamily member 1B; ANG: angiogenin; ANGPT1: angiopoietin 1; sVEGFR-1: soluble vascular endothelial growth factor receptor-1; VEGFA: vascular endothelial growth factor A; CYP19A1: cytochrome P450 family 19 subfamily A member 1; DUSP6: dual specificity phosphatase 6; ERα: estrogen receptor 1; PGR: progesterone receptor; STAR: steroidogenic acute regulatory protein.; n.e.: not expressed.

**Table 2 ijms-24-16678-t002:** Associations between urinary concentrations of PBs and BPs and expression levels of genes involved in cell adhesion (n = 22).

	ITGB2					CLDN7			
	β	95% CI		*p*-Value		β	95% CI		*p*-Value
** Parabens **									
**MeP**	** *0.30* **	** *0.11* **	** *0.49* **	** *0.004* **		** *0.17* **	** *0.02* **	** *0.32* **	** *0.026* **
*<39.26* ng/mL	0.00	-	-			0.00	-	-	
*>39.26* ng/mL	** *1.13* **	** *0.36* **	** *1.90* **	** *0.006* **		** *0.85* **	** *0.35* **	** *1.35* **	** *0.002* **
**EtP**	** *0.23* **	** *0.05* **	** *0.40* **	** *0.013* **		0.08	−0.06	0.21	0.260
*<4.51* ng/mL	0.00	-	-			0.00	-	-	
*>4.51* ng/mL	0.74	−0.13	1.61	0.091		−0.11	−0.93	0.71	0.780
**PrP**	0.04	−0.18	0.25	0.723		0.01	−0.14	0.16	0.869
*<2.54* ng/mL	0.00	-	-			0.00	-	-	
*>2.54* ng/mL	−0.01	−0.95	0.93	0.986		−0.22	−0.86	0.42	0.478
**BuP**	0.17	−0.19	0.53	0.343		0.15	−0.05	0.35	0.135
*<0.14* ng/mL	0.00	-	-			0.00	-	-	
*>0.14* ng/mL	** *1.07* **	** *0.05* **	** *2.09* **	** *0.040* **		** *0.91* **	** *0.21* **	** *1.62* **	** *0.014* **
**ΣPBs**	** *0.29* **	** *0.08* **	** *0.51* **	** *0.011* **		** *0.18* **	** *0.02* **	** *0.33* **	** *0.029* **
*<53.31* ng/mL	0.00	-	-			0.00	-	-	
*>53.31* ng/mL	** *0.85* **	** *0.01* **	** *1.70* **	** *0.049* **		** *0.67* **	** *0.11* **	** *1.23* **	** *0.022* **
** Benzophenones **									
**BP-1**	0.10	−0.45	0.66	0.697		−0.24	−0.53	0.05	0.101
*<1.42* ng/mL	0.00	-	-			0.00	-	-	
*>1.42* ng/mL	0.70	−0.39	1.78	0.848		0.06	−0.76	0.88	0.882
**BP-3**	0.04	−0.38	0.46	0.848		−0.16	−0.38	0.06	0.150
*<2.53* ng/mL	0.00	-	-			0.00	-	-	
*>2.53* ng/mL	0.48	−0.63	1.59	0.380		−0.10	−0.91	0.72	0.810
**4-OHBP**	0.06	−0.65	0.76	0.870		−0.13	−0.52	0.25	0.482
*<0.73* ng/mL	0.00	-	-			0.00	-	-	
*>0.73* ng/mL	−0.17	−1.30	0.96	0.754		0.12	−0.70	0.94	0.768
**ΣBPs**	0.07	−0.55	0.70	0.809		−0.28	−0.61	0.04	0.079
*<7.43* ng/mL	0.00	-	-			0.00	-	-	
*>7.43* ng/mL	0.63	−0.47	1.72	0.245		−0.16	−0.97	0.66	0.696

CI: confidence interval; MeP: methylparaben; EtP: ethylparaben; PrP: propylparaben; BuP: butylparaben; ΣPB: sum of parabens; BP-1: benzophenone-1; BP-3: benzophenone-3; 4-OHBP: 4-hydroxibenzophenone; ΣBP: sum of benzophenones; ITGB2: integrin beta-2; CLDN7: claudin 7.

**Table 3 ijms-24-16678-t003:** Associations between urinary concentrations of PBs and BPs and expression levels of genes involved in invasion, migration, and metastasis (n = 22).

	**MMP1**					**FUT8**					**RRM2**			
	**OR**	**95% CI**		***p*-Value**		**OR**	**95% CI**		***p*-Value**		**β**	**95% CI**		***p*-Value**
** Parabens **														
**MeP**	** *2.96* **	** *1.14* **	** *7.72* **	** *0.026* **		0.90	0.55	1.46	0.668		0.10	−0.094	0.303	0.283
*<39.26* ng/mL	1.00	-	-			1.00	-	-			0.00	-	-	
*>39.26* ng/mL	** *12.00* **	** *1.12* **	** *128.84* **	** *0.040* **		0.66	0.11	4.00	0.648		0.63	−0.10	1.36	0.088
**EtP**	1.27	0.83	1.95	0.277		1.17	0.77	1.76	0.466		0.09	−0.72	0.26	0.252
*<4.51* ng/mL	1.00	-	-			1.00	-	-			0.00	-	-	
*>34.51* ng/mL	3.75	0.54	26.05	0.181		3.75	0.54	26.05	0.181		0.41	−0.36	1.18	0.280
**PrP**	1.63	0.93	2.84	0.086		1.26	0.81	1.96	0.306		0.05	−0.14	0.23	0.609
*<2.54* ng/mL	1.00	-	-			1.00	-	-			0.00	-	-	
*>2.54* ng/mL	1.52	0.25	9.30	0.648		** *12.00* **	** *1.12* **	** *128.84* **	** *0.040* **		0.18	−0.61	0.96	0.647
**BuP**	1.46	0.73	2.19	0.285		0.67	0.33	1.35	0.262		0.03	−0.23	0.29	0.827
*<0.14* ng/mL	1.00	-	-			1.00	-	-			0.00	-	-	
*>0.14* ng/mL	5.00	0.70	35.50	0.108		1.17	0.19	7.12	0.867		0.71	−0.34	1.75	0.174
**ΣPBs**	** *2.68* **	** *0.09* **	** *6.59* **	** *0.031* **		0.93	0.56	1.52	0.760		0.10	−0.11	0.32	0.317
*<53.31* ng/mL	1.00	-	-			1.00	-	-			0.00	-	-	
*>53.31* ng/mL	** *12.00* **	** *1.12* **	** *128.84* **	** *0.040* **		0.66	0.11	4.00	0.648		0.33	−0.45	1.10	0.387
** Benzophenones **														
**BP-1**	0.59	0.22	1.55	0.281		1.41	0.56	3.54	0.467		0.077	−0.302	0.455	0.677
*<1.42* ng/mL	1.00	-	-			1.00	-	-			0.00	-	-	
*>1.42* ng/mL	0.66	0.11	4.00	0.648		1.52	0.25	9.30	0.648		0.44	−0.32	1.21	0.239
**BP-3**	1.23	0.63	2.39	0.542		1.06	0.54	2.09	0.872		−0.13	−0.42	0.15	0.343
*<2.53* ng/mL	1.00	-	-			1.00	-	-			0.00	-	-	
*>2.53* ng/mL	1.52	0.25	9.30	0.648		3.75	0.54	26.045	0.181		0.40	−0.67	1.48	0.443
**4-OHBP**	1.86	0.55	6.34	0.322		1.476	0.475	4.583	0.501		−0.13	−0.604	0.352	0.587
*<0.73* ng/mL	1.00	-	-			1.00	-	-			0.00	-	-	
*>0.73* ng/mL	1.52	0.25	0.30	0.648		1.52	0.25	9.30	0.648		−0.44	−1.20	0.32	0.243
**ΣBPs**	** *1.22* **	** *0.02* **	** *0.88* **	** *0.036* **		1.29	0.47	3.58	0.620		−0.07	−0.50	0.35	0.725
*<7.43* ng/mL	1.00	-	-			1.00	-	-			0.00	-	-	
*>7.43* ng/mL	0.66	0.11	4.00	0.648		1.52	0.25	9.30	0.648		0.55	−0.51	1.61	0.289
	**MDK**					**RHOB**					**SPRY2**			
	**OR**	**95% CI**		***p*-value**		**β**	**95% CI**		***p*-value**		**OR**	**95% CI**		***p*-value**
** Parabens **														
**MeP**	0.77	0.47	1.27	0.309		** *0.27* **	** *0.11* **	** *0.43* **	** *0.002* **		** *0.48* **	** *0.24* **	** *0.97* **	** *0.040* **
*<39.26* ng/mL	1.00	-	-			0.00	-	-			1.00	-	-	
*>39.26* ng/mL	0.45	0.08	2.67	0.379		** *1.02* **	** *0.08* **	** *1.96* **	** *0.035* **		0.31	0.05	1.85	0.200
**EtP**	0.88	0.60	1.31	0.529		** *0.18* **	** *0.03* **	** *0.32* **	** *0.022* **		0.89	0.60	1.30	0.535
*<4.51* ng/mL	1.00	-	-			0.00	-	-			1.00	-	-	
*>34.51* ng/mL	1.00	0.18	5.68	1.000		0.30	−0.76	1.35	0.565		1.46	0.26	8.05	0.665
**PrP**	0.94	0.62	1.42	0.772		0.11	−0.07	0.28	0.206		0.79	0.52	1.21	0.274
*<2.54* ng/mL	1.00	-	-			0.00	-	-			1.00	-	-	
*>2.54* ng/mL	1.00	0.18	5.68	1.000		−0.50	−1.84	0.83	0.441		0.69	0.12	3.78	0.665
**BuP**	0.72	0.38	1.36	0.308		** *0.40* **	** *0.11* **	** *0.69* **	** *0.010* **		0.63	0.34	1.16	0.137
*<0.14* ng/mL	1.00	-	-			0.00	-	-			1.00	-	-	
*>0.14* ng/mL	0.75	0.13	4.29	0.746		** *1.68* **	** *0.56* **	** *2.79* **	** *0.005* **		0.43	0.07	2.50	0.346
**ΣPBs**	0.76	0.45	1.27	0.293		** *0.28* **	** *0.12* **	** *0.45* **	** *0.002* **		** *0.51* **	** *0.26* **	** *0.98* **	** *0.044* **
*<53.31* ng/mL	1.00	-	-			0.00	-	-			1.00	-	-	
*>53.31* ng/mL	0.45	0.08	2.67	0.379		** *1.11* **	** *0.19* **	** *2.03* **	** *0.021* **		0.31	0.05	1.85	0.200
** Benzophenones **														
**BP-1**	0.71	0.30	1.70	0.438		−0.02	−0.42	0.38	0.923		1.03	0.45	2.35	0.954
*<1.42* ng/mL	1.00	-	-			0.00	-	-			1.00	-	-	
*>1.42* ng/mL	0.45	0.08	2.67	0.379		0.30	−1.06	1.65	0.653		0.69	0.12	3.78	0.665
**BP-3**	0.83	0.43	1.58	0.567		0.03	−0.36	0.42	0.865		0.83	0.42	1.61	0.572
*<2.53* ng/mL	1.00	-	-			0.00	-	-			1.00	-	-	
*>2.53* ng/mL	2.22	0.38	13.18	0.379		0.85	−0.45	2.15	0.187		1.46	0.26	8.05	0.665
**4-OHBP**	0.44	0.13	1.53	0.197		0.234	−0.267	0.735	0.340		0.70	0.24	2.05	0.510
*<0.73* ng/mL	1.00	-	-			0.00	-	-			1.00	-	-	
*>0.73* ng/mL	1.00	0.18	5.68	1.000		0.04	−1.32	1.40	0.947		0.69	0.12	3.78	0.665
**ΣBPs**	0.82	0.31	2.15	0.689		−0.01	−0.44	0.42	0.957		0.98	0.38	2.51	0.967
*<7.43* ng/mL	1.00	-	-			0.00	-	-			1.00	-	-	
*>7.43* ng/mL	0.45	0.08	2.67	0.379		0.85	−0.34	2.04	0.151		0.69	0.12	3.78	0.665

OR: odds ratio; CI: confidence interval; MeP: methylparaben; EtP: ethylparaben; PrP: propylparaben; BuP: butylparaben; ΣPB: sum of parabens; BP-1: benzophenone-1; BP-3: benzophenone-3; 4-OHBP: 4-hydroxibenzophenone; ΣBP: sum of benzophenones; MMP1: matrix metallopeptidase 1; FUT8: fucosyltransferase 8; RRM2: ribonucleotide reductase M2; MDK: midkine; RHOB: ras homolog gene family, member B; SPRY2: sprout homolog 2.

**Table 4 ijms-24-16678-t004:** Associations between urinary concentrations of PBs and BPs and expression levels of genes involved in inflammation (n = 22).

	IL1RL1					IL6ST					NR3C1					TNFRSF1B			
	β	95% CI		*p*-Value		β	95% CI		*p*-Value		β	95% CI		*p*-Value		OR	95% CI		*p*-Value
** Parabens **																			
**MeP**	−0.21	−0.77	0.35	0.440		** *0.35* **	** *0.12* **	** *0.58* **	** *0.005* **		**0.57**	**0.13**	**1.02**	**0.014**		0.81	0.47	1.37	0.426
*<39.26* ng/mL	0.00	-	-			0.00	-	-			0.00	-	-			1.00	-	-	
*>39.26* ng/mL	2.00	−2.32	6.31	0.345		** *1.26* **	** *0.38* **	** *2.13* **	** *0.007* **		1.09	−1.05	3.22	0.300		0.39	0.06	2.77	0.346
**EtP**	−0.12	−0.58	0.33	0.575		** *0.23* **	** *0.07* **	** *0.40* **	** *0.008* **		0.30	−0.07	0.67	0.107		0.99	0.65	1.49	0.950
*<4.51* ng/mL	0.00	-	-			0.00	-	-			0.00	-	-			1.00	-	-	
*>34.51* ng/mL	−1.32	−3.31	0.67	0.179		0.25	−0.80	1.31	0.620		−0.42	−2.61	1.77	0.692		1.00	0.15	6.53	1.000
**PrP**	−0.20	−0.69	0.29	0.391		0.05	−0.19	0.29	0.662		0.21	−0.21	0.63	0.305		1.02	0.66	1.59	0.929
*<2.54* ng/mL	0.00	-	-			0.00	-	-			0.00	-	-			1.00	-	-	
*>2.54* ng/mL	−1.31	−5.68	3.06	0.538		−0.42	−1.86	1.02	0.549		−0.69	−2.86	1.48	0.513		2.57	0.36	18.33	0.346
**BuP**	0.43	−0.20	1.06	0.165		** *0.47* **	** *0.20* **	** *0.73* **	** *0.002* **		** *0.74* **	** *0.27* **	** *1.21* **	** *0.004* **		0.42	0.15	1.18	0.101
*<0.14* ng/mL	0.00	-	-			0.00	-	-			0.00	-	-			1.00	-	-	
*>0.14* ng/mL	3.91	−0.08	7.91	0.055		** *1.71* **	** *0.49* **	** *2.94* **	** *0.008* **		** *2.61* **	** *0.78* **	** *4.44* **	** *0.007* **		0.30	0.04	2.17	0.232
**ΣPBs**	−0.25	−0.83	0.33	0.379		** *0.36* **	** *0.13* **	** *0.60* **	** *0.004* **		** *0.60* **	** *0.14* **	** *1.05* **	** *0.013* **		0.78	0.45	1.36	0.379
*<53.31* ng/mL	0.00	-	-			0.00	-	-			0.00	-	-			1.00	-	-	
*>53.31* ng/mL	2.29	−1.99	6.57	0.278		** *1.27* **	** *0.40* **	** *2.14* **	** *0.006* **		1.26	−0.85	3.37	0.225		0.39	0.06	2.77	0.346
** Benzophenones **																			
**BP-1**	−0.13	−1.19	0.93	0.797		−0.03	−0.50	0.44	0.886		−0.33	−1.26	0.59	0.459		0.58	0.21	1.62	0.298
*<1.42* ng/mL	0.00	-	-			0.00	-	-			0.00	-	-			1.00	-	-	
*>1.42* ng/mL	−1.39	−4.97	2.19	0.425		0.32	−1.13	1.77	0.649		−1.18	−4.74	2.39	0.499		2.57	0.36	18.33	0.346
**BP-3**	−0.17	−1.10	0.77	0.714		−0.03	−0.42	0.36	0.862		0.31	−0.48	1.09	0.428		** *10.09* **	** *1.34* **	** *76.29* **	** *0.025* **
*<2.53* ng/mL	0.00	-	-			0.00	-	-			0.00	-	-			1.00	-	-	
*>2.53* ng/mL	2.31	−2.45	7.06	0.324		0.60	−0.83	2.03	0.389		2.15	−1.32	5.62	0.211		** *11.00* **	** *1.05* **	** *120.40* **	** *0.049* **
**4-OHBP**	2.23	−0.57	5.03	0.112		−0.04	−0.69	0.61	0.895		0.31	−0.78	1.40	0.555		1.58	0.48	5.17	0.452
*<0.73* ng/mL	0.00	-	-			0.00	-	-			0.00	-	-			1.00	-	-	
*>0.73* ng/mL	** *4.71* **	** *0.36* **	** *9.06* **	** *0.035* **		0.17	−1.29	1.62	0.810		1.87	−1.63	5.38	0.278		1.00	0.15	6.53	1.000
**ΣBPs**	−0.15	−1.46	1.16	0.812		−0.13	−0.64	0.38	0.593		−0.35	−1.57	0.87	0.558		** *14.15* **	** *1.27* **	** *157.40* **	** *0.031* **
*<7.43* ng/mL	0.00	-	-			0.00	-	-			0.00	-	-			1.00	-	-	
*>7.43* ng/mL	1.87	−1.67	5.42	0.281		0.40	−1.04	1.84	0.569		0.54	−1.65	2.71	0.614		8.33	0.78	89.47	0.080

OR: odds ratio; CI: confidence interval; MeP: methylparaben; EtP: ethylparaben; PrP: propylparaben; BuP: butylparaben; ΣPB: sum of parabens; BP-1: benzophenone-1; BP-3: benzophenone-3; 4-OHBP: 4-hydroxibenzophenone; ΣBP: sum of benzophenones; IL1RL1: interleukin 1 receptor, type I; IL6ST: interleukin 6 cytokine family signal transducer; NR3C1: nuclear receptor subfamily 3 group C member 1; TNFRSF1B: tumor necrosis factor receptor superfamily member 1B.2.2.4. Markers of Angiogenesis.

**Table 5 ijms-24-16678-t005:** Associations between urinary concentrations of PBs and BPs and expression levels of genes involved in angiogenesis (n = 22).

	ANG					ANGPT1					sVEFGR-1					VEGFA			
	β	95% CI		*p*-Value		β	95% CI		*p*-Value		β	95% CI		*p*-Value		OR	95% CI		*p*-Value
** Parabens **																			
**MeP**	0.22	−0.01	0.46	0.062		** *0.49* **	** *0.18* **	** *0.81* **	** *0.004* **		1.53	0.87	2.67	0.139		0.88	0.53	1.44	0.602
*<39.26* ng/mL	0.00	-	-			0.00	-	-			0.00	-	-			1.00	-	-	
*>39.26* ng/mL	** *0.99* **	** *0.14* **	** *1.84* **	** *0.025* **		** *1.62* **	** *0.27* **	** *2.97* **	** *0.021* **		2.22	0.38	13.18	0.379		1.00	0.15	6.53	1.000
**EtP**	0.15	−0.05	0.35	0.141		0.22	−0.10	0.55	0.160		1.14	0.77	1.69	0.508		0.82	0.53	1.27	0.369
*<4.51* ng/mL	0.00	-	-			0.00	-	-			0.00	-	-			1.00	-	-	
*>34.51* ng/mL	0.35	−0.85	1.55	0.549		0.15	−1.43	1.73	0.842		2.22	0.38	13.18	0.379		0.12	0.01	1.29	0.080
**PrP**	0.14	−0.08	0.35	0.201		0.26	−0.09	0.60	0.132		** *1.86* **	** *1.01* **	** *3.40* **	** *0.045* **		0.85	0.54	1.33	0.465
*<2.54* ng/mL	0.00	-	-			0.00	-	-			0.00	-	-			1.00	-	-	
*>2.54* ng/mL	0.21	−1.00	1.41	0.722		0.38	−1.19	1.95	0.620		5.40	0.78	37.51	0.088		0.39	0.06	2.77	0.346
**BuP**	0.24	−0.02	0.51	0.071		−0.25	−1.81	1.30	0.736		1.15	0.64	2.073	0.650		0.95	0.51	1.75	0.865
*<0.14* ng/mL	0.00	-	-			0.00	-	-			0.00	-	-			1.00	-	-	
*>0.14* ng/mL	0.94	−0.20	2.07	0.100		−2.04	−6.69	2.61	0.371		** *7.50* **	** *1.04* **	** *54.12* **	** *0.046* **		0.50	0.07	3.55	0.488
**ΣPBs**	0.23	−0.01	0.48	0.063		** *0.51* **	** *0.19* **	** *0.84* **	** *0.004* **		1.54	0.87	2.74	0.140		0.89	0.53	1.49	0.664
*<53.31* ng/mL	0.00	-	-			0.00	-	-			0.00	-	-			1.00	-	-	
*>53.31* ng/mL	0.58	−0.60	1.76	0.320		** *1.78* **	** *0.49* **	** *3.07* **	** *0.010* **		2.22	0.38	13.18	0.379		1.00	0.15	6.53	1.000
** Benzophenones **																			
**BP-1**	0.05	−0.42	0.52	0.837		−0.55	−1.23	0.13	0.103		1.08	0.46	2.52	0.865		0.67	0.25	1.78	0.419
*<1.42* ng/mL	0.00	-	-			0.00	-	-			0.00	-	-			1.00	-	-	
*>1.42* ng/mL	0.68	0.48	1.84	0.240		0.48	−4.25	5.20	0.836		2.22	0.38	13.18	0.379		0.12	0.01	1.29	0.080
**BP-3**	0.03	−0.32	0.39	0.843		0.05	−0.60	0.70	0.879		0.76	0.37	1.56	0.453		0.55	0.22	1.41	0.215
*<2.53* ng/mL	0.00	-	-			0.00	-	-			0.00	-	-			1.00	-	-	
*>2.53* ng/mL	0.79	−0.36	1.94	0.169		0.53	−1.03	2.09	0.482		2.22	0.38	13.18	0.379		0.12	0.01	1.29	0.080
**4-OHBP**	−0.25	−0.83	0.34	0.388		−0.15	−1.18	0.89	0.767		1.47	0.48	4.55	0.501		1.16	0.35	3.77	0.810
*<0.73* ng/mL	0.00	-	-			0.00	-	-			0.00	-	-			1.00	-	-	
*>0.73* ng/mL	−0.16	−1.37	1.04	0.779		−0.03	−1.62	1.55	0.964		2.22	0.38	13.18	0.379		2.57	0.36	18.33	0.346
**ΣBPs**	0.03	−0.49	0.56	0.894		−0.43	−1.30	0.45	0.316		0.80	0.30	2.13	0.654		0.50	0.15	1.63	0.250
*<7.43* ng/mL	0.00	-	-			0.00	-	-			0.00	-	-			1.00	-	-	
*>7.43* ng/mL	0.47	−0.72	1.66	0.421		0.83	−2.55	4.20	0.613		1.00	0.18	5.68	1.000		0.39	0.06	2.77	0.346

OR: odds ratio; CI: confidence interval; MeP: methylparaben; EtP: ethylparaben; PrP: propylparaben; BuP: butlparaben; ΣPB: sum of parabens; BP-1: benzophenone-1; BP-3: benzophenone-3; 4-OHBP: 4-hydroxibenzophenone; ΣBP: sum of benzophenones; ANG: angiogenin; ANGPT1: angiopoietin 1; sVEGFR-1: soluble vascular endothelial growth factor receptor-1; VEGFA: vascular endothelial growth factor A.2.2.5. Markers of Cell Proliferation and Hormonal Stimulation.

**Table 6 ijms-24-16678-t006:** Associations between low and high urinary concentrations of PBs and BPs and expression levels of genes involved in cell proliferation and hormonal stimulation (n = 22).

	DUSP6					ERα					STAR			
	β	95% CI		*p*-Value		β	95% CI	*p*-Value			β	95% CI		*p*-Value
** Parabens **														
**MeP**	** *−0.90* **	** *−1.49* **	** *−0.30* **	** *0.005* **		** *0.18* **	** *0.04* **	** *0.31* **	** *0.015* **		−1.35	−2.908	0.21	0.086
*<39.26* ng/mL	0.00	-	-			0.00	-	-			0.00	-	-	
*>39.26* ng/mL	−3.13	−6.99	0.73	0.106		** *0.78* **	** *0.30* **	** *1.26* **	** *0.003* **		** *−5.64* **	** *−11.08* **	** *−0.20* **	** *0.043* **
**EtP**	0.01	−0.63	0.65	0.968		0.10	−0.11	0.21	0.074		−0.51	−1.84	0.83	0.437
*<4.51* ng/mL	0.00	-	-			0.00	-	-			0.00	-	-	
*>34.51* ng/mL	−1.74	−6.52	3.03	0.450		0.29	−0.31	0.87	0.325		0.02	−6.02	6.06	0.994
**PrP**	0.16	−0.56	0.89	0.640		0.06	−0.08	0.19	0.406		−0.07	−1.31	1.16	0.902
*<2.54* ng/mL	0.00	-	-			0.00	-	-			0.00	-	-	
*>2.54* ng/mL	−1.38	−6.18	3.42	0.560		0.05	−0.56	0.65	0.871		0.38	−5.66	6.42	0.898
**BuP**	−0.49	−1.65	0.68	0.391		0.16	−0.02	0.35	0.076		0.57	−1.25	2.39	0.520
*<0.14* ng/mL	0.00	-	-			0.00	-	-			0.00	-	-	
*>0.14* ng/mL	−3.01	−7.67	1.64	0.192		** *0.70* **	** *0.19* **	** *1.20* **	** *0.009* **		1.50	−4.53	7.52	0.610
**ΣPBs**	** *−0.83* **	** *−1.46* **	** *−0.19* **	** *0.014* **		** *0.19* **	** *0.05* **	** *0.33* **	** *0.011* **		−1.28	−2.82	0.27	0.100
*<53.31* ng/mL	0.00	-	-			0.00	-	-			0.00	-	-	
*>53.31* ng/mL	1.37	−8.24	0.81	0.102		** *0.39* **	** *0.21* **	** *1.21* **	** *0.008* **		−3.47	−9.29	2.35	0.228
** Benzophenones **														
**BP-1**	0.97	−0.28	2.21	0.122		−0.09	−0.35	0.17	0.463		** *2.94* **	** *0.38* **	** *5.50* **	** *0.027* **
*<1.42* ng/mL	0.00	-	-			0.00	-	-			0.00	-	-	
*>1.42* ng/mL	1.37	−3.43	6.18	0.558		0.39	−0.50	1.29	0.372		4.20	−1.51	9.92	0.140
**BP-3**	0.17	−0.85	1.18	0.737		−0.12	−0.33	0.10	0.262		0.41	−1.68	2.51	0.683
*<2.53* ng/mL	0.00	-	-			0.00	-	-			0.00	-	-	
*>2.53* ng/mL	−0.05	−4.90	4.79	0.982		−0.01	−0.92	0.91	0.988		2.45	−3.49	8.38	0.400
**4-OHBP**	−0.30	−2.00	1.40	0.713		0.02	−0.35	0.39	0.923		2.53	−0.92	5.98	0.141
*<0.73* ng/mL	0.00	-	-			0.00	-	-			0.00	-	-	
*>0.73* ng/mL	−0.47	−5.31	4.37	0.841		−0.01	−0.62	0.60	0.971		** *5.94* **	** *0.94* **	** *10.93* **	** *0.022* **
**ΣBPs**	0.85	−0.60	2.29	0.234		−0.17	−0.49	0.15	0.273		2.10	−0.84	5.03	0.151
*<7.43* ng/mL	0.00	-	-			0.00	-	-			0.00	-	-	
*>7.43* ng/mL	−0.58	−5.41	4.26	0.807		0.09	−0.82	1.01	0.832		4.61	−0.70	9.92	0.085

CI: confidence interval; MeP: methylparaben; EtP: ethylparaben; PrP: propylparaben; BuP: butylparaben; ΣPB: sum of parabens; BP-1: benzophenone-1; BP-3: benzophenone-3; 4-OHBP: 4-hydroxibenzophenone; ΣBP: sum of benzophenones; DUSP6: dual specificity phosphatase 6; ERα: estrogen receptor 1; STAR: steroidogenic acute regulatory protein.

## Data Availability

Data are contained within the article and Appendix A.

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
