# Peer review of "Expression Profiles of Genes Related to Development and Progression of Endometriosis and Their Association with Paraben and Benzophenone Exposure"

_ijms, 2023, doi:10.3390/ijms242316678_

Round 1
Reviewer 1 Report
Comments and Suggestions for Authors
Peinado et al, demonstrate a positive association between urinary paraben and benzophenone concentrations and the expression of genes involved in the development and progression of endometriosis in a cohort of patients with endometriosis at different stages of severity.
Table 1 has already been published in Peinado et al, Sci Total Environ 2023 (doi: 10.1016/j.scitotenv.2023.163014). It cannot be republished here.
Gene expression is assessed by RT-qPCR and expressed as 2-delta delta Ct. This approach requires an amplification efficiency of the reference gene approximately equal to those of the target genes of interest, and close to 100%. What are the efficiencies of the primers used (for all the genes studied)? Furthermore, this approach corresponds to the difference in Ct between the sample tested and a positive control for expression of the gene of interest. For each gene analyzed, it would be interesting to know which positive control was used.
Urine concentration results are expressed as an arithmetic mean. Why choose the arithmetic mean rather than the geometric mean or median?
The cohort included 33 women, but only 22 had analyzable samples for gene expression and urinary assay. The authors sometimes present results for all 33 women/samples (Table 1, Table 2, ...) and sometimes for all 22 samples (Table 3 to 7). This leads to confusion. Why not present results for only those women for whom tissue and urine were available?
Why was a threshold of 75% detection chosen for the genes considered in the rest of the study?
The authors often conclude that the genes studied are overexpressed. However, we don't know what this overexpression is compared to, as this study did not include patients without endometriosis. Conversely, it would have been very interesting and informative to assess whether there is an association between the stage of severity of the pathology, gene expression and urinary paraben or benzophenone concentrations.
Minor revision:
- A sentence setting out the context of the study would be appreciated in the abstract.
- Line 50 : what does PCP mean?
- Line 58 : the formatting of the reference is not identical to the rest of the manuscript
- Table 2-7: What does mean represent? What is the unit?
Reviewer 2 Report
Comments and Suggestions for Authors
The presented manuscript entitled “Expression profiles of genes related to development and progression of endometriosis and their association with paraben 3 and benzophenone exposure” presents very interesting results, however, in this form could not be published. The main issues of the present studies are poor set of applied analyses. In my opinion, toe use of only one method, qPCR, should be enriched by the protein studies of chosen genes products to confirm the Authors theses and validate the obtained results. The Authors investigated large number of genes connected with many processes, whereas, in my opinion, they should focus more on genes and their products involved in one or two processes and investigate them more complex. For example, the Authors, beside checking the expression of StAR and CYP19A1 should also design a Western blot experiment to confirm this and enrich this by the in vitro studies confirming the influence of the investigated factors on the E2 secretion. Another example, in the case of angiogenesis and proposed influence on signaling pathways, the Authors should investigate the influence on the phosphorylation of proteins involved in the mentioned pathway. It is, in my opinion, very important since in mammals the correlation between genes and their protein products expression may be very low (even 30%), and the presented study are not the full transcriptomic studies involving the use of NGS or microarray method.
Another issue is very poor described methodology of qPCR experiments. Please support more specific information about primers. Were they designed by the Authors or the sequences were taken from another publications? If the Authors designed them by themselves, therefore they should describe the used software, as well as the results of validation experiment. Please support more information to the table about the reference id of the sequences they were working with, as well as primers sequences and concentration. There is also a lack of information about qPCR reaction conditions for each gene. Today there is a common and practice to normalize the gene expression with the use not one reference gene, but with the use of the geometric mean of expression of at least 2 reference genes. Please detail this paragraph adhering to MIQUE guidelines (doi: 10.1373/clinchem.2008.112797). Please also attach the information did the authors confirmed the reaction specificity by the analysis of melting curve. Please also attach the reference to deltaCt method (Livak & Smittgen 2001).
Please also modify the headlines in the Results section. Since the authors did not investigate the process, but only the markers, they should rename the chapters for example for “the markers of angiogenesis” etc.
Since the Authors did not investigate the expression of downstream elements of signaling pathways, I would avoid such statements as ‘signaling pathway’ in the Discussion section.
There is also lack of information about funding and the number of Ethical Committee approval.
Comments on the Quality of English LanguageNo complaints.
Reviewer 3 Report
Comments and Suggestions for Authors
This is an innovative and well-written manuscript on the expression levels of 23 genes involved in cell signaling pathways including cell adhesion, invasion, inflammation, angiogenesis, and proliferation in women with endometriosis and their expression levels association with EDCs (BPs and PBs) exposure. I would like to suggest some minor comments to improve its quality.
The authors should avoid duplicating words used in their keywords and the title of the manuscript.
The authors should cite the relevant references (lines 61-63).
I would suggest adding some sentences about biomarkers of endometriosis in the introduction section. This article can be helpful Biomarkers of endometriosis - ScienceDirect.
Reviewer 4 Report
Comments and Suggestions for Authors
Peinado et al. report a series of association analyses between EDC exposures and endometriotic lesion expression levels of genes related cell adhesion, invasion/migration, inflammation, angiogenesis, and cell proliferation/hormone stimulation. They identify statistically significant associations between several parabens and benzophenones and genes that could be plausibly linked to endometriosis disease etiology and/or progression. This study provides valuable clinical data that could be relevant to linking environmental exposures to endometriosis. The study suffers from low sample numbers, a limited number of genes analyzed, and lack of a clear link between changes in gene expression with clinical correlates. However, limitations are acknowledged by the authors, and these data may be useful in designing future mechanistic studies and larger clinical studies, making this study valuable for the research community after some minor modifications.
General comments:
11. The clinical relevance of correlations between PBs/BPs and gene expression levels is weak when the majority of the genes tested were not associated with any differences in disease stage or symptoms (only FUT8 and SPRY2 genes were significantly correlated with disease stage). Are there any other metrics by which the genes tested could be directly tied to clinical outcomes in this dataset? This would make the findings much more biologically interesting. As it stands, any clinical impact of the findings is only theoretical.
22. Dot plots to visually display correlations between PBs/BPs and particular genes (in addition to the tables) would be beneficial for more transparent data reporting and conceptual understanding on the part of the reader.
33. Why was this particular panel of PBs and BPs chosen for this study? Are there any known differences between the individual EDCs that might explain their differential correlations with the genes analyzed in this study? Please add to the Introduction or Discussion as appropriate.
Specific comments:
11. In-text citation #15 is formatted inconsistently as superscript (line 58)
22. The methods section (line 127) states that RT-PCR primer sequences are detailed in Supplementary Table S1, but the table only shows assay ID’s from the supplier. Please either report the sequences or correct the manuscript text.
33. Most of the time this manuscript uses the term “endometriotic lesion” but instead uses the term “endometrioma” in Table 1 and Table S2. If there is a justification for this, it should be explained. Otherwise, the more general term “endometriotic lesion” should be used in all cases to avoid confusion.
44. Should BuP be short for “butylparaben”? Tables 3-7 and S3 define it as “buthylparaben”.
55. Line 258 should state “Supplementary Table S4” and “Supplementary Table S5”.
66. In all relevant tables except Table 3, the lines under EtP read “<4.51 ng/mL” and “>34.51 ng/mL”. If Table 3 is correct, “>34.51” should be “>4.51”.
